# Single-fluorophore membrane transport activity sensors with dual-emission read-out

Cindy Ast[1], Roberto De Michele[2], Michael U Kumke[3], Wolf B Frommer[1]*

[1]Department of Plant Biology, Carnegie Institution for Science, Stanford California, United States; [2]Institute of Biosciences and Bioresources, Italian National Research Council, Palermo, Italy; [3]Department of Physical Chemistry, Institute of Chemistry, University of Potsdam, Potsdam, Germany

**Abstract** We recently described a series of genetically encoded, single-fluorophore-based sensors, termed AmTrac and MepTrac, which monitor membrane transporter activity in vivo (*De Michele et al., 2013*). However, being intensiometric, AmTrac and Meptrac are limited in their use for quantitative studies. Here, we characterized the photophysical properties (steady-state and time-resolved fluorescence spectroscopy as well as anisotropy decay analysis) of different AmTrac sensors with diverging fluorescence properties in order to generate improved, ratiometric sensors. By replacing key amino acid residues in AmTrac we constructed a set of dual-emission AmTrac sensors named deAmTracs. deAmTracs show opposing changes of blue and green emission with almost doubled emission ratio upon ammonium addition. The response ratio of the deAmTracs correlated with transport activity in mutants with altered capacity. Our results suggest that partial disruption of distance-dependent excited-state proton transfer is important for the successful generation of single-fluorophore-based dual-emission sensors.

*For correspondence: wfrommer@carnegiescience.edu

## Main text

Quantitative analysis of transport activity in vivo is extremely challenging, and popular methods, such as crystallography, patch clamping and radioactivity assays can only give limited insights since they are often destructive or are restricted to cell surface studies. We recently engineered single fluorescent protein (FP)-based ammonium transport activity sensors, termed AmTrac and MepTrac, which allowed monitoring of the activity of ammonium transporters in living yeast cells by means of fluorescence intensity (FI) measurements (*De Michele et al., 2013*). Both AmTrac and MepTrac are composed of a circularly permuted enhanced green fluorescent protein (cpEGFP) inserted into the central loop of plant or yeast AMT/MEP ammonium transporters. Quantitative fluorescence spectroscopy of yeast cells expressing the sensor variants demonstrates that they accurately report transporter activity as 30–50% FI decrease of green FI (*De Michele et al., 2013*). Being based on a single FP, AmTrac and MepTrac are intensiometric, since the readout consists of FI modulations of a single wavelength. However, ratiometric signal output would be clearly advantageous for sensor calibration and reliable quantitative imaging (*Tantama et al., 2012*). Only a few existing single FP sensors are ratiometric, displaying two emission maxima at a single excitation wavelength. The pH-sensors deGFP1 and deGFP4 (*Hanson et al., 2002*) and the calcium sensor GEM-GECO1 (*Zhao et al., 2011*) belong to this category. Based on similar steady-state fluorescence spectra of AmTrac and deGFP1 and deGFP4 (*Hanson et al., 2002*), we reasoned that it should be possible to turn AmTrac into a ratiometric, dual-emission sensor.

In AmTrac, the cpEGFP was inserted between residue 233 and 234 of the *Arabidopsis thaliana* ammonium transporter AtAMT1;3 (**Figure 1A**, top). The AmTrac sensors differ only by two residues at the N-terminus of the cpEGFP, also referred to as left linker, the right linker being composed of phenylalanine (F) and asparagine (N) in all sensors. The first AmTrac engineered, also named AmTrac-LE, carries leucine (L, Leu[234]) and glutamate (E, Glu[235]) in the left linker. Brighter versions carry a serine (S, Ser[235]) in the second position of the left linker, and variable amino acids in the first position. AmTrac-GS carries a glycine (G, Gly[234]), AmTrac-LS a leucine (L, Leu[234]) and AmTrac-IS an isoleucine (I, Ile[234]) (**Figure 1A**).

Despite the minimal differences in their amino acid sequence, AmTrac-GS, -LS, -IS and LE differ greatly in the shape and intensities of their fluorescence spectra, when expressed in yeast (**Figure 1B**). The fluorescence spectra of all AmTracs show two excitation maxima, at $\lambda_{exc} \sim 390$ nm and $\lambda_{exc} \sim 500$ nm,

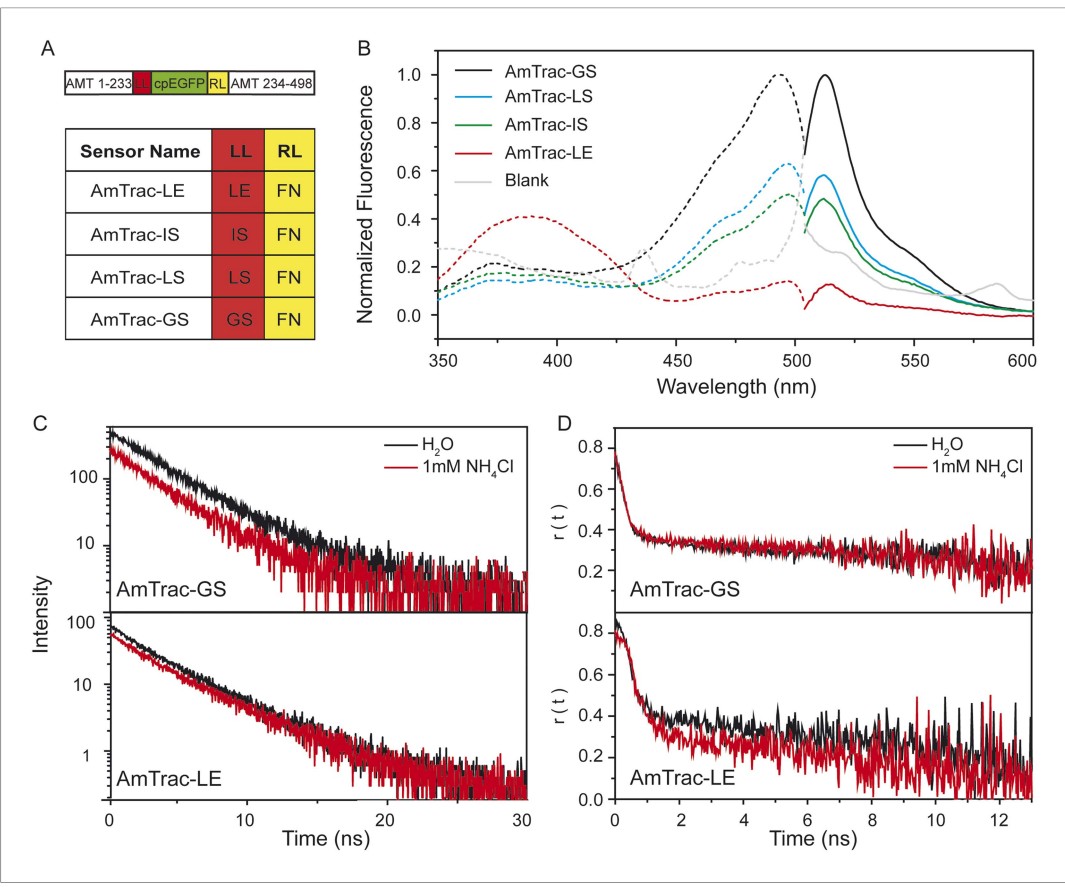

**Figure 1**. Steady-state and time-resolved fluorescence analysis of AmTrac sensors. (**A**) Scheme of AmTrac sensors showing left (LL) and right linker (RL) positions (top). Table indicates amino acid composition of linkers in AmTrac sensors. (**B**) Normalized fluorescence spectra, excitation (dashed lines; $\lambda_{em}$ 514 nm), and emission (solid line; $\lambda_{exc}$ 485 nm). Traces in grey represent blank (background fluorescence of untransformed yeast). Values relate to major peak of AmTrac-GS (n = 3). (**C**) Time-resolved fluorescence decay kinetics of AmTrac-GS and -LE in presence and absence of 1 mM NH$_4$Cl ($\lambda_{exc}$ 475 nm; $\lambda_{em}$ 514 nm; n = 3). (**D**) Time-resolved fluorescence anisotropy r(t) of AmTrac-GS and -LE in presence and absence of 1 mM NH$_4$Cl ($\lambda_{exc}$ 475 nm; $\lambda_{em}$ 514 nm).

The following figure supplements are available for figure 1:

**Figure supplement 1**. Summary of lifetime components ($\tau$) and $\chi^2$ values obtained for AmTrac-GS and -LE treated with water as control or 1 mM NH$_4$Cl.

**Figure supplement 2**. Time-resolved fluorescence decay kinetics of AmTrac-LE in presence and absence of 1 mM NH$_4$Cl ($\lambda_{exc}$ 475 nm, $\lambda_{em}$ 514 nm).

also referred to as A- and B-band hereafter. In AmTrac-LE, the A-state dominates the fluorescence spectrum, whereas all other AmTracs (-GS, -LS and -IS) show an equilibrium shift toward the B-state. In wild type GFP (wtGFP) (44), the A-band derives from the A-state, the protonated chromophore, while the B-state indicates the deprotonated chromophore (43). Both states are coupled by a protonation equilibrium in both wtGFP and the AmTrac sensors.

The generally accepted mechanism of single-FP sensors assumes changes in the solvent accessibility and consequent quenching of the fluorescence (*Akerboom et al., 2009*; *Chen et al., 2013*). However, AmTracs are substantially different from the soluble sensors characterized so far. Membrane localization of the sensors reduces motion and mobility. Therefore, we performed time-resolved fluorescence measurements to detect if non-radiative processes such as solvent quenching or homo-FRET are involved in the sensor response. This analysis was performed for AmTrac-GS and -LE (*Figure 1C,D*). AmTrac-GS showed single exponential decay kinetics with lifetime values of $\tau = 2.55 \pm 0.04$ ns and $\tau = 2.48 \pm 0.1$ ns when treated with water and ammonium, respectively (*Figure 1C* top and *Figure 1—figure supplement 1*). These values are in agreement with most reports on a single lifetime of about $\tau = 2.5$ to 2.8 ns for EGFP in solution (*Stepanenko et al., 2004*; *Arpino et al., 2012*). AmTrac-LE showed bi-exponential decay kinetics with fluorescence decay times of $\tau_1 = 1.96 \pm 0.4$ ns and $\tau_2 = 3.98 \pm 0.32$ ns and $\tau_1 = 1.69 \pm 0.07$ ns and $\tau_2 = 3.96 \pm 0.17$ ns when treated with water or ammonium, respectively (*Figure 1C* bottom and *Figure 1—figure supplements 1, 2*). The two lifetime components found for AmTrac-LE are indicative of the presence of two species in the excited state. We thus conclude that the response mechanism of AmTrac-LE vs -GS differs with regard to which state (A- or B-state) is excited, and postulate that either different excited state reactions and/or different excited state species are involved. Since the lifetime values in absence and presence of ammonium differed, at most, by less than 1 ns for AmTrac-LE and -GS (*Figure 1—figure supplement 1*) we exclude external quenching effects due to altered solvent access in AmTracs, which is the widely accepted mechanism of single-FP sensors.

The distance of the cpEGFP moieties in the sensors allows for homo-FRET, but anisotropy decay kinetics changed only minimally in the presence of ammonium for AmTrac-GS and -LE (*Figure 1D*). The large initial r-values found are most likely artificial and probably due to weak signal of the green fluorescence and strong light scattering of the yeast cells. We thus conclude that the FI decrease is not the result of a change in the efficiency of (a possible) homo-FRET and postulate that structural rearrangements affect the protonation states and thus the FI.

Excitation of the A-state of wtGFP leads to green emission due to excited-state proton transfer (ESPT) (*Chattoraj et al., 1996*). ESPT in wtGFP is a photophysical phenomenon that describes the rapid deprotonation of the neutral chromophore in the excited state leading to loss of blue fluorescence and formation of a green-emitting intermediate state (I-state). The proton travels from the chromophore hydroxyl group, via a buried water molecule and neighboring side chain serine 205 (Ser[205]), to the final acceptor Glu[222] after excitation of the A-form. ESPT is usually prevented in EGFP due to the S65T mutation (*Brejc et al., 1997*).

To test for ESPT in AmTracs, we recorded fluorescence emission spectra of yeast transformed with AmTrac-LE and -GS after excitation of the A-state ($\lambda_{exc}$ 395 nm), in presence of varying ammonium concentrations. Green fluorescence was detected, further supporting the hypothesis of ESPT occurrence (*Figure 2B* and *Figure 2—figure supplement 1A*). However, AmTrac-LE failed to fully respond to ammonium treatment when the A-state was excited, since the FI decreased by 30% only (*Figure 2B*). Conversely, when the B-state was excited, the FI decreased up to ~50% (*Figure 2A*). The emission spectrum of AmTrac-LE was characterized by another unique feature: the excitation of the A-state showed two emission maxima, one higher energy shoulder-like maximum at $\lambda_{em}$ 490 nm and a distinct peak at $\lambda_{em}$ 515 nm. Upon ammonium addition, the relative intensity of the peaks was altered and an iso-emissive point was observed (*Figure 2B*). Interestingly, this iso-emissive behavior was detected for AmTrac-LE but not for AmTrac-GS (*Figure 2—figure supplement 1A*).

Crystal structures of AmTrac sensors are not available yet. However, crystal structures have been solved for cpEGFP, as free protein or within sensors (*Wang et al., 2008*; *Akerboom et al., 2009*; *Leder et al., 2010*) and can provide clues on how the linker residues are involved in the structural rearrangements during the sensing mechanism. The crystal structure of cpEGFP (PDB 3EVP) shows two rotamers for Glu[148] in close proximity to the chromophore Thr65–Tyr66–Gly67, which indicates a high degree of flexibility at position 148. We assume that due to re-arrangements in the cpEGFP barrel, Glu[148] may exist in different orientations and distances to the phenyl-group of the

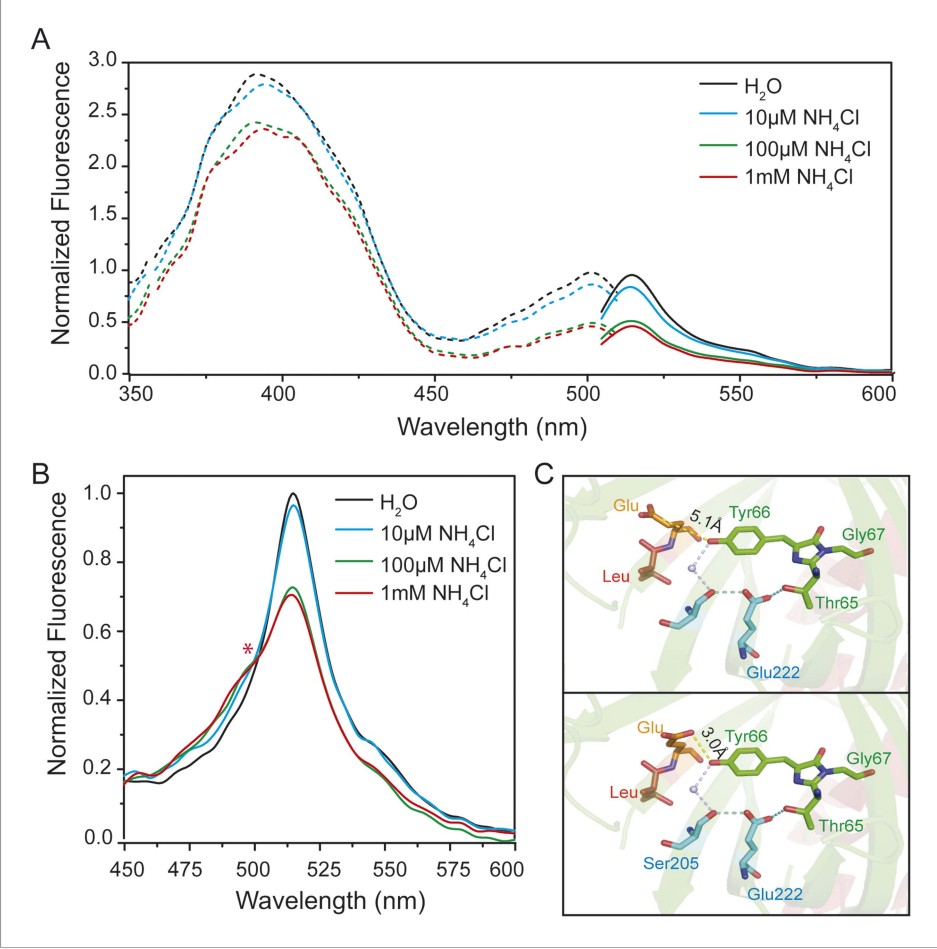

**Figure 2**. Steady-state fluorescence spectra for excited-state proton transfer (ESPT) analysis of yeast transformed with AmTrac-LE and postulated structural differences. (**A**) Fluorescence excitation (dashed line; $\lambda_{em}$ 530 nm) and emission (solid line; $\lambda_{exc}$ 485 nm) after treatment with NH$_4$Cl at indicated concentrations. Values were normalized to the major peak of the B-band (n = 3). (**B**) Fluorescence emission spectrum ($\lambda_{exc}$ 395 nm). Values normalized to major maximum of the water-treated control (n = 3). Asterisk (*) indicates iso-emissive point. (**C**) Chromophore environment of crystalized circularly permuted enhanced green fluorescent protein (cpEGFP, PDB 3EVP). Chromophore is depicted in green; residues involved in ESPT via a buried water molecule in blue, left linker residues in orange and red. Top and bottom illustrations show different rotamers of glutamate Glu[148] with indicated distances of hydrogen bonds to chromophore.

The following figure supplement is available for figure 2:

**Figure supplement 1**. Steady-state fluorescence spectrum for ESPT analysis of intact yeast transformed with AmTrac-GS (n = 3) and postulated structural differences.

chromophore (*Figure 2C*). Since Glu[148] corresponds to Glu[235] in AmTrac-LE, we consider the solved structure of cpEGFP as a proxy for AmTrac-LE. Theoretical modifications of the linker sequence were performed using Pymol by replacing the linker residues LE of cpEGFP (PDB 3EVP) with GS. The result shows only one conformational state for serine of the GS linker (*Figure 2—figure supplement 1B*) indicating more stability during conformation changes. This also indicates the impact of different amino acid linkers on the structural environment of the chromophore. We postulate that Glu[235] of AmTrac-LE relocates during the sensing process, most likely altering the efficiency of ESPT and in turn leading to blue and green fluorescence emission.

The spectral behavior of AmTrac-LE shows similarities with the dual-emission pH sensor deGFP1 (*Hanson et al., 2002*) as well as the GFP mutant S65T/H148E (*Shu et al., 2007*). Therefore, we reasoned

that it should be possible to generate improved dual-emission sensors with a larger ratio change compared to AmTrac-LE. We modified the left linker amino acids only, since they appeared to be involved in the dual-emission behavior. We replaced LE with two random amino acids, to check for all possible combinations, and analyzed these constructs in yeast. Out of more than 500 yeast colonies screened, we found four candidates with the new linker composition CP, FP, RP, YP that displayed distinct dual-emission patterns and two of these candidates showed improved ratios upon ammonium treatment. Interestingly, all four constructs, named deAmTracs, contained a proline residue at the linker position of the former Glu[235], while the first amino acid of the linker varied (*Figure 3* and *Figure 3—figure supplement 1*). The two candidates with the highest ratio changes, deAmTrac-CP and -FP, are described in more detail (*Figure 3*). deAmTrac-CP carried a cysteine and proline, while deAmTrac-FP contained phenylalanine and proline in the left linker. Upon excitation with $\lambda_{exc}$ 395 nm, blue and green emission maxima at 490 nm and 515 nm, respectively, were detected, which showed opposing changes in FI upon ammonium treatment. A clear iso-emissive point in the dual-emission spectra was observed (*Figure 3A*). Plotting the response ratio of the intensity of the A-state /I-state vs the external ammonium concentration showed that the ratio values almost doubled for both deAmTrac-CP and -FP (*Figure 3B*). The affinity constants from the response ratio towards ammonium for deAmTrac-CP ($EC_{50}$ 54 ± 9 μM) and deAmTrac-FP ($EC_{50}$ 36 ± 6 μM) were similar to AmTrac ($EC_{50}$ 55 ± 7 μM) (*De Michele et al., 2013*).

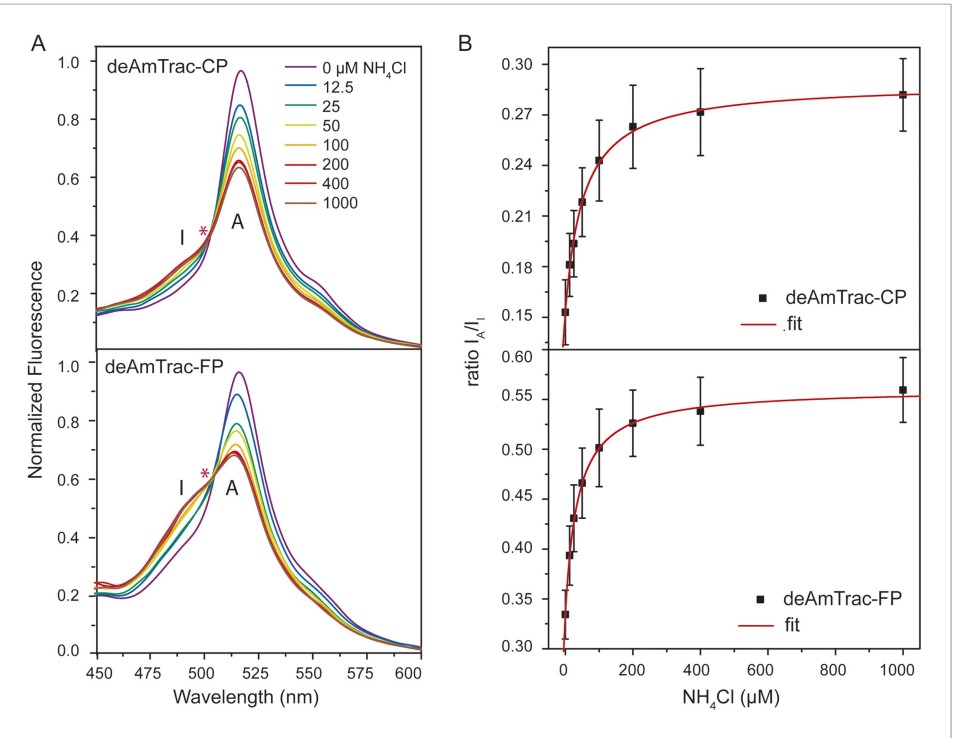

**Figure 3**. Steady-state fluorescence titration analysis of intact yeast transformed with deAmTrac-CP and -FP. (**A**) Fluorescence emission spectrum ($\lambda_{exc}$ 395 nm) recorded after NH$_4$Cl addition (concentrations indicated). Values normalized to maximum of water treated control. Asterisks (*) indicate the iso-emissive point. A refers to the protonated A-state, I is the deprotonated intermediate I-state of the chromophore. (**B**) Titration of the fluorescence response at indicated NH$_4$Cl concentrations plotted as ratio of the intensity of the A-state /I-state ($I_A/I_I$) (mean ± SE; n = 5).

The following figure supplements are available for figure 3:

**Figure supplement 1**. Summary of properties of AmTrac sensor variants in response to water or 1 mM NH$_4$Cl with A-state excition ($\lambda_{exc}$ = 395–400 nm).

**Figure supplement 2**. Chromophore environment of crystalized cpEGFP (PDB 3EVP).

Due to the appearance of blue emission we speculate that proline or glutamic acid at position 235 are involved in hindering deprotonation or stabilizing the protonated state of the chromophore. Theoretical modification of the LE linker in the crystal structure of cpEGFP (PDB 3EVP) for either CP or FP using Pymol yields different rotamers (*Figure 3—figure supplement 2*). We postulate that the residues CP and FP allow for more flexibility due to the possibility of different conformational states. Differently from GS in AmTrac-GS, these linker residues may partially disrupt the ESPT pathway, preventing rapid proton transfer and trapping the excited A-state which consequently emits blue light. The blue emission is rather small, since the green emission from successful ESPT dominates the spectrum.

Several lines of evidence demonstrated that AmTrac's fluorescence response is strictly correlated to the ammonium transporter activity (*De Michele et al., 2013*). To verify that such correlation was maintained in deAmTracs, we generated a series of mutants with altered transporter activity, based on the suppressor screen for the inactivating mutation T464D in AmTrac-LE (*De Michele et al., 2013*). In deAmTrac-CP and -FP, the T464D mutation was still sufficient to block growth and response. In the T464D mutant background the presence of the suppressing mutation A141E increased the ratiometric response and transport capacity, shown as reduced growth of the mutants on high ammonium concentrations. Conversely, mutation Q61E restored growth and response of deAmTrac-T464D, although not to the full extent (*Figure 4—figure supplement 1*). Overall, the mutations had an effect on transport activity and response similar to AmTrac-LE (*De Michele et al., 2013*) (*Figure 4A* and *Figure 4—figure supplement 1*). Spectral analysis of the mutant deAmTracs showed the impact of the introduced mutations on the dual-emission pattern. In the T464D mutant the blue emission disappeared, and only green emission was detected. In presence of A141E, blue emission increased and green emission decreased. Q61E shows the opposite effect with decreased blue and increased green emission at high ammonium concentrations (*Figure 4B*). These results support the utility of deAmTracs in assessing ammonium transporter activity. However, our findings also point out that such sensors are complex systems and that a single mutation affecting the structure and function of the transporter may also affect the direct environment of the chromophore.

Based on spectral analyzes and the ESPT model reported by *Chattoraj et al. (1996)*, we postulate a mechanistic model with three co-existing processes: (i) structural rearrangements in the sensor protein upon ammonium perception, (ii) a predominantly protonated state of the cpEGFP chromophore and (iii) partial disruption of the ESPT pathway to prevent efficient proton transfer. In the absence of ammonium transport activity, an efficient hydrogen bond network is established in close proximity to the chromophore, allowing for fast proton transfer from the excited state of the neutral chromophore to the final acceptor. Under these conditions, green emission from the deprotonated chromophore is observed. With increasing ammonium transport activity, the well-established ESPT pathway is partially inhibited, so that the proton cannot be efficiently transferred. Emission from the protonated intermediate is observed as blue light (*Figure 5*).

The backbone rearrangements must be extremely fast to disrupt and re-establish the ESPT network. Still, the motion is not dramatic enough to completely prevent ESPT, explaining why green emission is still observed as the dominant peak during conditions of high ammonium transport. Therefore, deAmTracs only show modest ratio changes compared to other published dual-emission sensors, possibly due to only small spatial rearrangements or averaging of the ensembles during the ammonium binding/transport process. Future efforts will address and explore alternative approaches for the design of improved ratiometric sensors for ammonium transporters.

## Materials and methods

### DNA constructs and mutagenesis

All cpEGFP-based sensor plasmid constructs described are based on pDRF'-AmTrac-LE and -GS; described previously (*De Michele et al., 2013*).

### Random mutagenesis

The deAmTrac sensors were generated by substituting the LE-linker of AmTrac-LE with random amino acids to test all the possible linker sequences and effects on dual-emission patterns. The cpEGFP fragment was amplified with forward primer AmXX FW containing the variable sequence encoding the

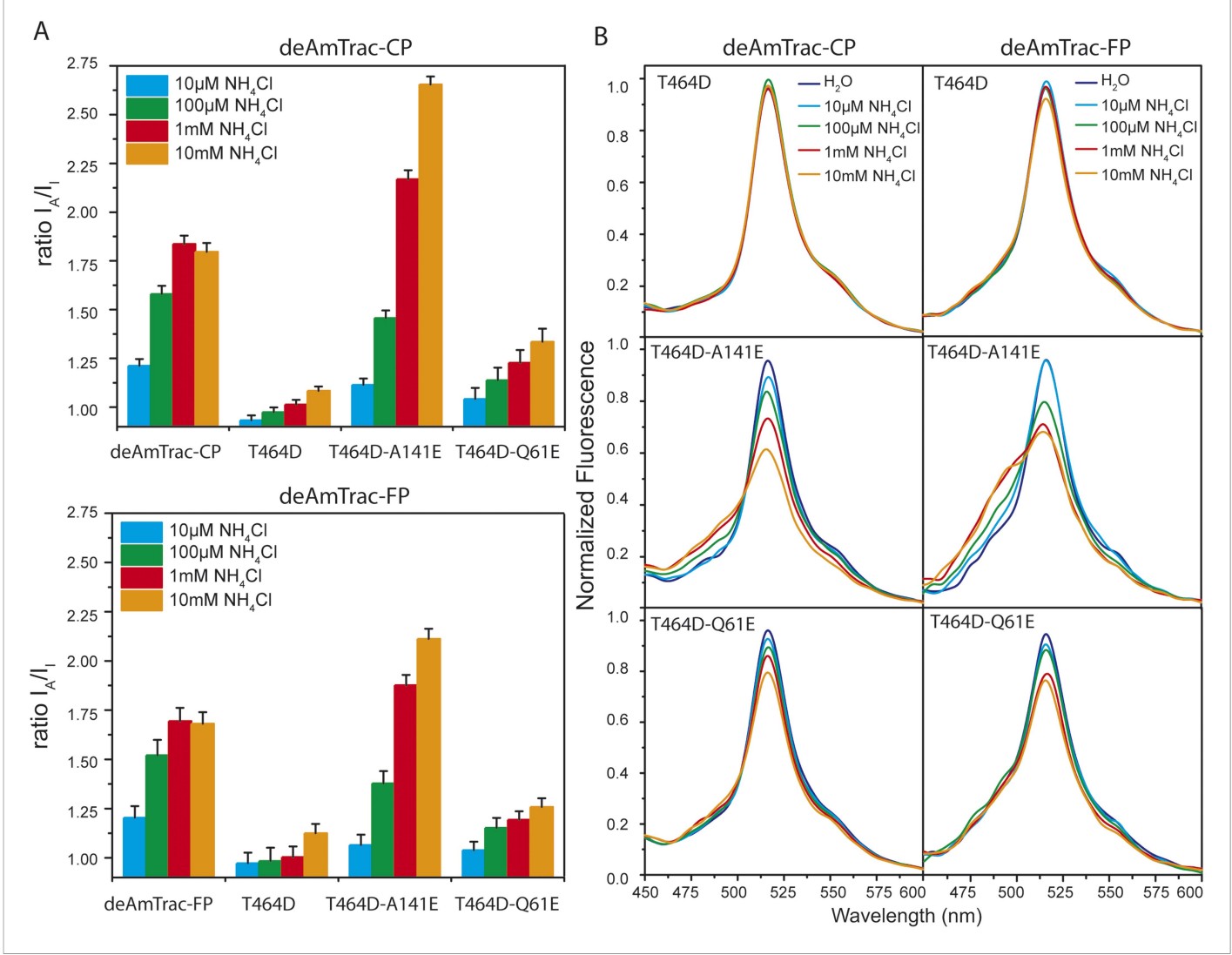

**Figure 4**. Response ratio and spectral analysis of mutant deAmTrac-CP and -FP. (**A**) Fluorescence response to $NH_4Cl$ (indicated concentrations) plotted as ratio of the intensity of the A-state /I-state ($I_A/I_I$). Data were normalized to water treated control (=1) (mean ± SE; n = 5). (**B**) Fluorescence emission spectra with $\lambda_{exc}$ 395 nm recorded after treatment with $NH_4Cl$ (concentrations indicated). Values are normalized to the maximum of the water treatment (n = 3).

The following figure supplement is available for figure 4:

**Figure supplement 1**. Yeast growth assay of *Δmep1,2,3* strain (31019b) transformed with mutant deAmTracs.

linker, and the reverse primer AmFN RV, encoding the C-terminal FN-linker (*Table 1*). DNA-fragments were gel-purified (Machery-Nagel) and yeast was co-transformed with linearized pDRF'-AtAMT1;3 (cleaved after amino acid 233). The cpEGFP-fragment with varying linkers was inserted after amino acid 233 of AtAMT1;3 by homologous recombination (*De Michele et al., 2013*).

## Site-directed mutagenesis

Site-directed mutagenesis was performed to generate deAmTrac-FP and -CP mutants. First, T464D was introduced using primers T464D-FW and T464D-RV (*Table 1*). Constructs deAmTrac-FP-T464D and deAmTrac-CP-T464D served as template to introduce a second mutation. Second mutations were either Q61E or A141E (primers Q61E FW and -RV, or A414E FW and -RV, respectively; *Table 1*).

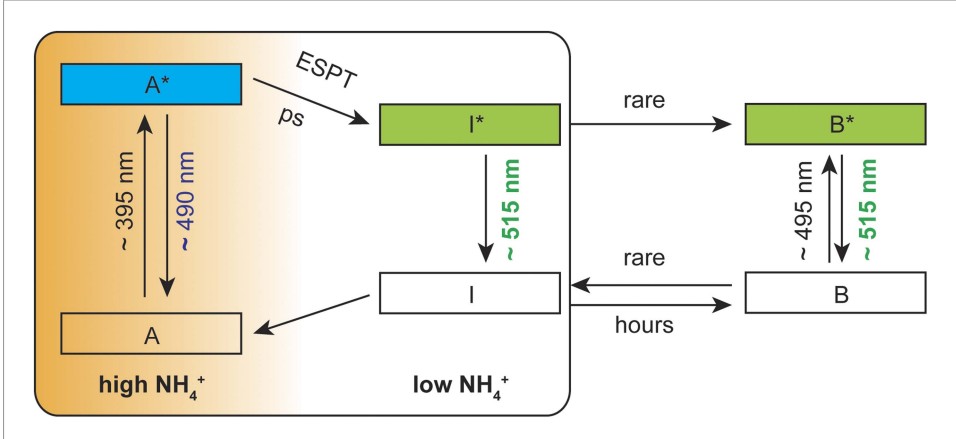

**Figure 5**. Suggested mechanistic ESPT model to describe the dual-emission in AmTrac-LE and deAmTracs during high and low ammonium transport activity. A, protonated A-state of the chromophore; I, deprotonated intermediate state; B, deprotonated B-state. Asterisks (*) indicate the excited state upon illumination with the indicated wavelength.

## Yeast transformation and culture

Measurements were carried out using yeast strain 31019b (*mep1Δ mep2Δ*::LEU2 *mep3Δ*::KanMX2 *ura3*), lacking endogenous MEP ammonium transporters (*Marini et al., 1997*). Yeast transformation was performed with lithium acetate (*Schiestl and Gietz, 1989*). Transformants were plated on solid YNB (minimal yeast medium without amino acids/ammonium sulfate; Difco BD, Franklin Lakes, NJ) supplemented with 3% glucose and 1 mM arginine. Single colonies were selected and inoculated in 5 ml liquid YNB supplemented with 3% glucose and 0.1% proline under agitation (230 rpm) at 30°C until $OD_{600nm}$ 0.5–0.9.

For the complementation assay, liquid cultures were diluted $10^{-1}$, $10^{-2}$, $10^{-3}$, $10^{-4}$ and $10^{-5}$ in water and 5 µl of each dilution was spotted on solid YNB medium buffered with 50 mM MES/Tris, pH 5.2, supplemented with 3% glucose and either $NH_4Cl$, or 1 mM arginine as the sole nitrogen source. After 3 days of incubation at 30°C, cell growth was documented by flatbed scanning the plate at 300 dpi in grayscale mode.

For fluorescence measurements, liquid yeast cultures were washed twice in 50 mM MES pH 6.0, and resuspended to $OD_{600nm}$ ~0.5 in 50 mM MES pH 6.0, supplemented with 5% glycerol to delay cell sedimentation (*Ast et al., 2015*).

## Steady-state fluorescence measurements of AmTrac-LE, -LS, -IS and -GS

Steady-state fluorescence of washed liquid cultures expressing AmTrac-LE, -LS, -IS and -GS (*Figure 1B*) was acquired using a Fluoromax-P fluorescence spectrometer (Horiba Jobin Yvon, Kyoto,

**Table 1**. List of mutagenesis primers

| Primer name | Primer sequence |
| --- | --- |
| AmXX FW | GAAGGTCCTCGTCGTGGTCGGTTCGAGAAANNNNNNAACGTCTATATCAAGGCCG |
| AmFN RV | CCGCGCAGAGCAATAGCGCGACCACC ATTAAA GTTGTACTCCAGCTTGTGCC |
| T464D FW | GCA AGG GAT GGA TAT GGA TCG TCA CGG TGG CTT TGC |
| T464D RV | GCA AAG CCA CCG TGA CGA TCC ATA TCC ATC CCT TGC |
| Q61E FW | CCT TGT CTT CGC CAT GGA GCT CGG CTT CGC TAT GC |
| Q61E RV | GCA TAG CGA AGC CGA GCT CCA TGG CGA AGA CAA GG |
| A141E FW | CAA TGG GCG TTC GCA ATC GAG GCC GCT GGA ATC AC |
| A141E RV | GTG ATT CCA GCG GCC TCG ATT GCG AAC GCC CAT TG |

Japan) and 3.5 ml silica cuvettes (Hellma Analytics, Mullheim, Germany). 2 ml cultures were diluted with 0.5 ml water before recording excitation/emission spectra ($\lambda_{em}$ = 514 nm, $\lambda_{exc}$ = 485 nm, step size 1 nm, n = 5). Untransformed yeast cells served as blank. Measurements were repeated independently at least three times.

## Time-resolved emission measurements of AmTrac-LE, and -GS

A FL920 fluorescence lifetime spectrometer (Edinburgh Instruments, Livingston, UK) operating in time-correlated single-photon counting mode was used for time-resolved emission measurements of yeast cultures expressing AmTrac-LE and -GS after treatment with 1 mM ammonium chloride. Green fluorescence was excited by a supercontinuum laser (SC-400-PP, Fianium, Southampton, UK) at $\lambda_{exc}$ = 475 nm at a rate of 20 MHz. Emission was detected at 90° relative to incoming beam. A 500 nm long pass filter (AHF Analysentechnik, Tübingen, Germany) in front of the detection system cut off scattered excitation light from the detector. AmTrac-GS was excited by vertically polarized light; the emission polarizer was set to magic angle (54.7°) to eliminate polarization from fluorescence decay. AmTrac-LE was measured without polarizers. Fluorescence decays were recorded for 5 min and analyzed with FAST software (Edinburgh Instruments, UK).

Fluorescence decays used to calculate the time-resolved anisotropy decay r(t) were recorded with vertically polarized excitation light and detected with emission polarizers set to vertical ($FI_V(t)$) or horizontal ($FI_H(t)$) positions, respectively. Anisotropy decay rates were calculated using *Equation 1* ($FI(t)$, FI at time t after laser pulse excitation at a given polarizer orientation; vertical v or horizontal h; G, instrument correction factor). G-factor was determined by measuring emission intensity with emission polarizers set to horizontal and vertical positions after horizontally polarized excitation.

$$r(t) = \frac{FI_V(t) - G*FI_H(t)}{FI_V(t) + 2G*FI_H(t)}. \tag{1}$$

## Fluorometric measurements for sensor response analysis

AmTrac-LE and -GS and deAmTrac variants were investigated using a fluorescence plate reader (Safire; Tecan, Männedorf, Switzerland). 200 µl of washed cells were loaded into black 96-well microplates with clear bottom (Greiner Bio-One, Germany) and treated with 50 µl ammonium solutions (10 mM, 1 mM, 100 µM or 10 µM $NH_4Cl$) or water as control. For the ammonium titrations, ammonium concentrations of 1000 µM, 400 µM, 200 µM, 100 µM, 50 µM, 25 µM, 12.5 µM or water were used. Cells were incubated for 8 min to saturate the response.

Fluorescence spectra were recorded in bottom reading mode using 7.5 nm bandwidth for both excitation and emission wavelengths ($\lambda_{exc}$ = 395 or 485 nm; $\lambda_{em}$ = 530 nm) and the values for single point fluorescence analysis were extracted ($\lambda_{em}$ = 490 nm or 515 nm). A minimum of three independent transformants was analyzed. The fit of the titration kinetics was performed using a dose–response function (*Equation 2*) (a, $EC_{max}$; b, $EC_{50}$; c, $EC_{baseline}$).

$$f(x) = \frac{a*x}{b + x} + c. \tag{2}$$

Graphs were created using OriginPro 8.6 G software (OriginLab, Northampton, MA, USA).

## Acknowledgements

We thank Soeren Gehne (University of Potsdam, Germany) for his help with the data analysis of the time-resolved fluorescence measurements and Lily S Cheung (Carnegie Institution for Science) for her help with the kinetics analysis and critical reading.

## Additional information

### Funding

| Funder | Grant reference | Author |
| --- | --- | --- |
| National Science Foundation (NSF) | MCB-1021677 | Wolf B Frommer |

| Funder | Grant reference | Author |
|---|---|---|
| National Science Foundation (NSF) | MCB-1413254 | Wolf B Frommer |

The funder had no role in study design, data collection and interpretation, or the decision to submit the work for publication.

## Author contributions

CA, MUK, Conception and design, Acquisition of data, Analysis and interpretation of data, Drafting or revising the article; RDM, WBF, Conception and design, Analysis and interpretation of data, Drafting or revising the article

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
