## [Decision Letter]

Thank you for sending your work entitled “Single-Fluorophore Membrane Transport Activity Sensors with Dual-Emission Read-Out” for consideration at *eLife*. Your article has been favorably evaluated by Detlef Weigel (Senior editor), Richard Aldrich (Reviewing editor), and two reviewers, and the two reviewers (Janice Robertson and Jonathan Javitch) have agreed to share their identity.

The Reviewing editor and the reviewers discussed their comments before we reached this decision, and the Reviewing editor has assembled the following comments to help you prepare a revised submission.

This is an interesting following up story to the original publication on a fluorescence-based ammonium binding/transport sensor, AmTrac. The authors carried out additional fluorescence characterization, which revealed a prominent “A-band” excitation in one of the sensors. They randomly modified one of the linkers and identified several new sensors with enhanced emission in the blue spectrum in response to ammonium. The most improved sensor has nearly a two-fold change in blue to green emission in response to high ammonium. This relatively modest change compared with dual emission sensors will limit the utility of the new sensors but the study nevertheless does provide interesting new information and a theoretical proposal for the underlying mechanism. The authors propose that small linker-related perturbations of excited-state proton transfer underlie the enhancement of the blue relative to the green emission, but argue that the small conformational changes aren't sufficient to create a more dramatic shift. This is an exciting development pushing forward the technology to directly observe membrane protein transport via fluorescent methods.

The following issues should be addressed in a revised version.

1) The study seems to have omitted a key piece of analysis: a full titration of the ratiometric fluorescence signal vs. external ammonium concentration, i.e. a quantification of the changes observed in Figures 2 and 3. This type of plot would demonstrate the range of sensitivity for the different constructs and clarify the number of states that are observed in the fluorescent read-out.

2) In addition, the paper could be strengthened by showing a series of tests of the ratiometric fluorescence signal, for instance, with different functional mutants that change the transport activity in a predicted manner. These types of experiments were elegantly carried out in the previous paper (7) and really showed the robustness of the fluorescence intensity changes on the original AmTrac sensors.

3) In the final sentence the authors write: “However, since most single-fluorophore sensors are based on the evaluation of a single fluorescence wavelength, this work will aid the design of future ratiometric single fluorophore sensors.” It is not clear how this will work. It is interesting that a disruption in ESPT may create ratiometric sensors, how to use this knowledge to rationally improve the sensor further is unclear as the authors have already tried 500 random mutations. We understand the urge to generalize the utility of the findings, but if the authors have some idea of a rational approach to extend this would be nice to hear it.

4) On several occasions the authors refer to theoretical modification or replacement of residues in the structure and show new conformations and distance measurements, but the methodology used was unclear. Is this simply visualizing based on the structure with a new side chain present or was some sort of molecular dynamics or energy minimization used?

---

## [Author Response]

*1) The study seems to have omitted a key piece of analysis: a full titration of the ratiometric fluorescence signal vs. external ammonium concentration, i.e. a quantification of the changes observed in*
Figures 2 and 3*. This type of plot would demonstrate the range of sensitivity for the different constructs and clarify the number of states that are observed in the fluorescent read-out*.

The reviewers raise an interesting concern. We added the titration curves of deAmTrac-CP and –FP together with the titration spectra in Figure 3.

*2) In addition, the paper could be strengthened by showing a series of tests of the ratiometric fluorescence signal, for instance, with different functional mutants that change the transport activity in a predicted manner. These types of experiments were elegantly carried out in the previous paper (*[7]*) and really showed the robustness of the fluorescence intensity changes on the original AmTrac sensors*.

The reviewers brought up a great suggestion. From the information obtained from [7] we chose 3 mutations as test cases for the sensor response and coupling between activity and fluorescence response. The T464D mutation inactivates the transporter activity and, consequentially, the sensor response. Two mutations suppressed this inactivation, the high capacity mutation A141E, which we expected to increase the ratio of the response, and one of the mutations with reduced transport activity Q61E, which we expected to show a decreased response ratio (Figure 4). The data not only support our hypothesis, but also the final mechanistic model in Figure 4, where low and high transport activities are correlated with ESPT.

*3) In the final sentence the authors write: “However, since most single-fluorophore sensors are based on the evaluation of a single fluorescence wavelength, this work will aid the design of future ratiometric single fluorophore sensors.” It is not clear how this will work. It is interesting that a disruption in ESPT may create ratiometric sensors, how to use this knowledge to rationally improve the sensor further is unclear as the authors have already tried 500 random mutations. We understand the urge to generalize the utility of the findings, but if the authors have some idea of a rational approach to extend this would be nice to hear it*.

We think that the insights obtained, including the occurrence of ESPT and presence of proline in the linker sequence, are useful to guide future efforts in designing ratiometric single fluorophore sensors. However, we agree with the reviewers that our results cannot be generalized. The results of deAmTrac-FP and –CP mutants show the complexity of these sensor systems. We have come to the conclusion that screening of randomly generated constructs will most likely be more successful. Thus we removed that sentence.

*4) On several occasions the authors refer to theoretical modification or replacement of residues in the structure and show new conformations and distance measurements, but the methodology used was unclear. Is this simply visualizing based on the structure with a new side chain present or was some sort of molecular dynamics or energy minimization used*?

The methodology was simple visualization using an existing structure of cpEGFP (3EVP) and replacing the side chains to illustrate the possibility of different conformations of certain residues. We did not attempt to predict different rotamers, rather to postulate a higher degree of flexibility of the linkers in the deAmTracs.